# Evaluating *de Novo* Assembly and Binning Strategies for Time Series Drinking Water Metagenomes

Solize Vosloo,[a] Linxuan Huo,[a] Christopher L. Anderson,[b] Zihan Dai,[c] Maria Sevillano,[b] Ameet Pinto[a]

[a]School of Civil and Environmental Engineering, Georgia Institute of Technology, Atlanta, Georgia, USA
[b]Department of Civil and Environmental Engineering, Northeastern University, Boston, Massachusetts, USA
[c]Research Center for Eco-Environmental Sciences, Chinese Academy of Sciences, Beijing, China

**ABSTRACT** Reconstructing microbial genomes from metagenomic short-read data can be challenging due to the unknown and uneven complexity of microbial communities. This complexity encompasses highly diverse populations, which often includes strain variants. Reconstructing high-quality genomes is a crucial part of the metagenomic workflow, as subsequent ecological and metabolic inferences depend on their accuracy, quality, and completeness. In contrast to microbial communities in other ecosystems, there has been no systematic assessment of genome-centric metagenomic workflows for drinking water microbiomes. In this study, we assessed the performance of a combination of assembly and binning strategies for time series drinking water metagenomes that were collected over 6 months. The goal of this study was to identify the combination of assembly and binning approaches that result in high-quality and -quantity metagenome-assembled genomes (MAGs), representing most of the sequenced metagenome. Our findings suggest that the metaSPAdes coassembly strategies had the best performance, as they resulted in larger and less fragmented assemblies, with at least 85% of the sequence data mapping to contigs greater than 1 kbp. Furthermore, a combination of metaSPAdes coassembly strategies and MetaBAT2 produced the highest number of medium-quality MAGs while capturing at least 70% of the metagenomes based on read recruitment. Utilizing different assembly/binning approaches also assists in the reconstruction of unique MAGs from closely related species that would have otherwise collapsed into a single MAG using a single workflow. Overall, our study suggests that leveraging multiple binning approaches with different metaSPAdes coassembly strategies may be required to maximize the recovery of good-quality MAGs.

**IMPORTANCE** Drinking water contains phylogenetic diverse groups of bacteria, archaea, and eukarya that affect the esthetic quality of water, water infrastructure, and public health. Taxonomic, metabolic, and ecological inferences of the drinking water microbiome depend on the accuracy, quality, and completeness of genomes that are reconstructed through the application of genome-resolved metagenomics. Using time series metagenomic data, we present reproducible genome-centric metagenomic workflows that result in high-quality and -quantity genomes, which more accurately signifies the sequenced drinking water microbiome. These genome-centric metagenomic workflows will allow for improved taxonomic and functional potential analysis that offers enhanced insights into the stability and dynamics of drinking water microbial communities.

**KEYWORDS** drinking water, longitudinal (time series) dataset, genome-resolved metagenomics

Address correspondence to Ameet Pinto, ameet.pinto@ce.gatech.edu.

Advances in high-throughput sequencing technologies have enabled characterization of microbial communities without the need for cultivation (1). This has greatly facilitated our understanding of microbial communities that inhabit a range of natural and engineered ecosystems. Two high-throughput sequencing technologies commonly

used to characterize microbial communities includes gene-targeted assays that use universal genes/regions (i.e., 16S rRNA, 18S rRNA, and internal transcribed spacer region for bacteria/archaea, eukaryotes, and fungi, respectively) and short-read shotgun DNA sequencing (i.e., metagenomics) (1–4). Other emerging sequencing approaches includes synthetic- and single-molecule long-read sequencing for both gene-targeted and metagenomic assays (5, 6). Gene-targeted assays provide valuable insights into the compositional and structural profiles of microbial communities in a fast and cost-effective manner; however, this approach is limited by challenges related to primer selection and amplification bias (7). Furthermore, taxonomic classification in gene-targeted assays is based on a fragment of a singular conserved universal marker gene that permits little resolution beyond the genus level and does not allow for the direct analysis of a microbial community's metabolic capabilities (8). In some instances, putative functional assignment is possible when using gene-targeted assays; however, this requires the availability of curated taxonomic databases and classification beyond the genus level (9). Limitations of gene-targeted assays can be overcome by utilizing genome-resolved metagenomics (10, 11). Genome-resolved metagenomics encompasses *de novo* assembly of short high-throughput paired-end reads into longer contiguous sequences (contigs) and subsequent reconstruction of metagenome-assembled genomes (MAGs) through clustering (or binning) of contigs based on nucleotide composition and differential coverage (12, 13). This approach offers improved taxonomic and functional potential analysis, as well as the characterization of novel microorganisms using phylogenetic analysis (14).

*De novo* assembly and reconstruction of MAGs from short-read metagenomic data can be challenging due to sequencing errors, repeats, depth of sequencing coverage, and the presence of strain variants (15, 16). These challenges influence the performance of assemblers, as it creates unresolved ambiguities in the reconstructed contigs, leading to erroneous and/or fragmented assemblies. Reconstructing high-quality MAGs is a crucial part of the genome-centric metagenomic workflow, as subsequent taxonomic, metabolic, and ecological inferences depend on the accuracy, quality, and completeness of genomes. Studies have attempted to optimize the recovery of high-quality assemblies and MAGs by benchmarking metagenomic software for assembly, binning, and taxonomic classification (16–19) as well as integrating metagenomic software in a modular workflow (i.e., MetaWRAP) and utilizing multiple binning algorithms (i.e., DAS Tool) to optimize the recovery of nonredundant, high-quality genomes (19, 20). However, owing to the unknown complexity of various environmental sample types, systematic evaluation of metagenomic workflows is required, as tool selection depends on the complexity of the biological sample and the availability of computational recourses (17, 21).

Genomes are often reconstructed by assembling all the samples together (coassembly) or creating individual assemblies. Coassembly is a computationally intensive approach that involves the pooling of multiple metagenomes, which allows for greater sequence depth and coverage as well as leveraging differential coverage of microorganisms across genomes for genome binning. While this assembly approach can facilitate the identification of populations that are present at lower abundances, it can also result in ambiguous and/or fragmented assemblies when strain-level variability is high (22, 23). In contrast, single-sample assemblies are computationally less intensive and are often used to reconstruct genomes of larger data sets and to preserve strain variation between different samples (24). It has also been shown that single-sample assembly produces more nonredundant high-quality MAGs and enables the reconstructions of genomes with similar phylogenetic placement to coassembled genomes (14). However, lower sequence depth and, thus, lower coverage resulting from single-sample assembly, in addition to the lack of differential coverage information, make genome reconstruction difficult when using this assembly approach, as coverage heuristics that are used to accurately disentangle repetitive sequences and differentiate between strain variants cannot be properly applied.

Microbial ecosystems within the urban water cycle are of relevance, given their impact on infrastructure as well as environmental and public health (25). Advances in our understanding of the drinking water microbiome have been greatly facilitated by the application of genome-resolved metagenomics (26–29). Drinking water systems (DWS) consist of diverse and complex indigenous microbial communities, with cell concentrations ranging between $10^3$ to $10^5$ cells/ml (30). Drinking water microbial communities are constantly adapting to environmental change and are influenced by seasonal fluctuations, water chemistry (i.e., disinfectant residuals), and infrastructure (i.e., pipe materials) (26, 31–34). From an environmental and public health perspective, insight into the compositional profile of microbial communities inhabiting DWS is of particular importance given their association with processes like nitrification that affect the quality of drinking water (27) and emerging contaminants (i.e., antibiotic-resistant genes [ARGs]) (35, 36) and proliferation of opportunistic pathogens that affect its safety (37–39).

Establishing a genome-centric metagenomic workflow for DWS is important to accurately characterize microbial species and elucidate their ecological relevance and functional potential. DWS harbors several log units fewer cells than other aquatic ecosystems (e.g., surface water and wastewater) (30), which makes it a difficult matrix for genome-resolved metagenomics, specifically because DNA yields are low, which is attributed to lower sequence depths that influence assembly and binning software in genome-centric metagenomic workflows. In addition to this, inferences of microbial community dynamics in DWS have been restrained by the limited availability of longitudinal metagenomic data sets, as most previous work was done utilizing gene-targeted assays in studies that were short (i.e., few time points), gapped (i.e., missing time points), and/or implemented over multiple spatiotemporal scales (31, 32, 40). Longitudinal data sets are preferred over cross-sectional studies (1), as they offer unique insights into the stability and dynamics of microbial communities. This is because information leveraged from these data sets can reveal periodic patterns that can be used in predictive modeling, describe irregularities in response to abrupt environmental perturbations, and capture temporal variation of microbial interactions (41).

Currently, there is little work on how to best leverage the unique properties of time series metagenomic data for DWS. Thus, the overall objective of this study was to evaluate the performance of a combination of *de novo* assembly and binning algorithms for time series metagenomic data for drinking water microbial communities. Our goal was to identify an ideal combination of assembly and binning strategies that can allow for high-quality metagenomic assemblies and MAGs that maximally capture the sequenced metagenomes. To evaluate the performance of the tested genome-centric metagenomic workflows, we utilized different measures of quality at the assembly and MAG levels. To assess the performance of assembly strategy and assembler, measures of contiguity (i.e., total assembly size, maximum contig length, $N_{50}$, $L_{50}$, etc.), gene calling and quality (i.e., coding DNA sequence [CDS], coding density), mapping rate, and rate of gene fragmentation and misassembly were used (21). We used the number of medium- to high-quality MAGs and proportion of metagenomic data retained in these MAGs to evaluate the performance of the combination of assembler/assembly strategy and binning approaches.

## RESULTS AND DISCUSSION

**Summary of metagenomic sequencing of drinking water samples.** On average, 23.03 $\pm$ 9.57 ng DNA was extracted from the 1,500-ml filtered tap water samples harboring between 21.8 and 85.8 million cells (Table S1 in the supplemental material). A total of 1.05 billion (mean $\pm$ standard deviation [SD], 87.67 $\pm$ 4.34 million reads) raw 150-nucleotide (nt) paired-end reads, ranging between 81.37 and 94.15 million reads per sample, were generated from the DNA extracts of 12 samples, which had average 16S rRNA gene counts of 3.8 $\times$ $10^5$ $\pm$ 1.9 $\times$ $10^5$ copies/$\mu$l (Table S3). Control samples with average 16S rRNA gene counts of 4.1 $\times$ $10^1$ $\pm$ 8 $\times$ $10^1$ copies/$\mu$l had at least

$3 \times 10^2$-fold less raw paired-ended reads than samples (0.27 $\pm$ 0.12 million reads) (Table S3). Processing of the raw paired-end reads following quality filtering and contaminant exclusion removed, on average, 1.02 $\pm$ 0.23% of the reads per sample. The final sequence data set consisted of 1.04 billion (86.75 $\pm$ 0.42 million) high-quality processed reads with a lower and upper range of 80.51and 93.08 million reads per sample, respectively (Table S3). Nonpareil (42) was used to assess the coverage of sequencing effort (Table S4). The average coverage estimates across samples were 89.00 $\pm$ 3.00%, with a lower and upper range of 84.00 and 94.00%, respectively. This suggests that a sequencing depth of ~81 to 94 million reads was sufficient to capture most of the microbial diversity in each sample.

**Evaluation of metagenome assembly quality for variable assembly strategy and metagenome assembler combinations.** The performance of two de Bruijn graph-based assemblers, metaSPAdes and MEGAHIT, that utilize iterative multiple k-mer approaches to improve assembly quality (43, 44) were assessed for three coassembly strategies (i.e., coassembly of all samples, MASH distance-based assembly, and time-discrete assembly) and assembly of individual samples (Table S2). Inclusion of various assembly strategies allows for the assessment of assembly performance in terms of computational requirements (i.e., RAM usage, assembly runtime per processing core, etc.) and assembly quality at various levels of diversity as well as sequence depth and coverage. The metaSPAdes assemblies required more computing recourses, i.e., demanded higher memory limits and threads and had runtimes that were up to 6-fold longer compared to the MEGAHIT assemblies (Table S5), which confirms previous findings (17, 18, 21). As expected, coassembly strategies (i.e., coassembly of all samples, MASH distance-based assembly, and time-discrete assembly) for both metaSPAdes and MEGAHIT were associated with longer runtimes (Table S5). Among the metaSPAdes assemblies, time-discrete assembly had the longest runtime (195 h summed across all assemblies), followed by coassembly (100 h) and MASH-distance based assembly (28 h summed across all assemblies). Similar observations were made for the MEGAHIT assemblies (Table S5).

Evaluating de novo assembly quality for environmental samples is challenging due to the lack of a ground truth reference assembly for comparison (45). As a result, we used measures of contiguity (i.e., total assembly size, maximum contig length, $N_{50}$, $L_{50}$, etc.), gene calling and quality (i.e., coding DNA sequence [CDS], coding density), mapping rate, and rate of gene fragmentation and misassembly to assess the quality of the assemblies (Table S5). In total, between 9,959,586 and 114,386,414 contigs were generated across the metaSPAdes and MEGAHIT assemblies. Most of the assembled contigs ~98.39% had lengths below 1 kbp and were not used in downstream analysis. Duplicate and contained contigs that accounted for between 10 and 42% of the filtered contigs (>1 kbp) were found among the MASH distance-based and time-discrete coassemblies and individual assemblies of metaSPAdes and MEGAHIT and removed (Table S5). Duplicate contigs were defined as contigs sharing 100% sequence similarity over the entire length, while contained contigs included shorter contigs that were 100% similar to a longer contig over their length. The average total number of contigs per assembly strategy kept after removing contigs shorter than 1 kbp and redundant contigs was 506,898 (SD, 164,157). For each assembly strategy, the metaSPAdes assemblies produced between 10 and 20% more contigs greater than 1 kbp than the number of contigs that were generated from the MEGAHIT assemblies (Table S5). Differences between assemblies were more apparent when metrics related to assembly contiguity were compared. Irrespective of the assembly strategy, the total assembly length of the metaSPAdes assemblies was greater than the assemblies of MEGAHIT (Fig. 1A). MetaSPAdes time-discrete assembly had the greatest assembly length (2,940.15 Mbp), followed by individual assembly (2,037.97 Mbp), coassembly (1,488.27 Mbp), and MASH distance-based assembly (1,147.06 Mbp). Since larger assembly lengths are not always indicative of better assembly quality (21), $N_{50}$ estimates representing a weighted medium contig size were considered. The metaSPAdes assemblies generated contigs with higher $N_{50}$ estimates than the MEGAHIT assemblies (Fig. 1B). Time-discrete

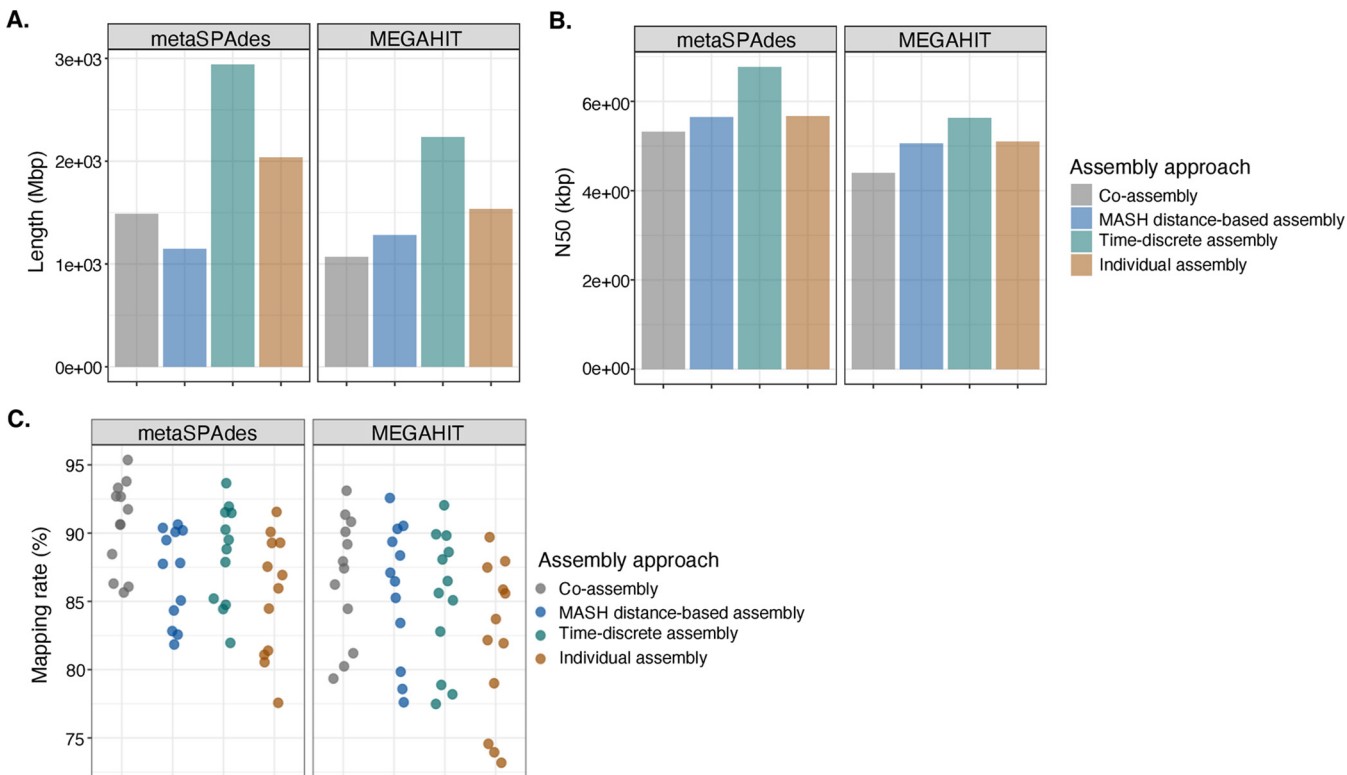

**FIG 1** Comparison of assembly characteristics associated with 4 different assembly strategies (coassembly [gray], MASH distance-based assembly [blue], time-discrete assembly [green], and individual assembly [orange]) that were assembled with metaSPAdes and MEGAHIT. Assembly characteristics were determined using nonredundant contigs larger than 1 kbp of coassemblies and pooled MASH distance-based, time-discrete, and individual assemblies. Assembly characteristics included total assembly length (A), $N_{50}$ estimates (B), and proportion of reads (C) of 12 drinking water samples (●) that were mapped against the nonredundant filtered assemblies. For a complete list of estimates, please refer to Table S5 in the supplemental material.

assembly of metaSPAdes had the highest $N_{50}$ estimates (6.77 kbp), followed by individual assembly (5.67 kbp), MASH distance-based assembly (5.65 kbp), and coassembly (5.32 kbp). The higher $N_{50}$ estimates of the metaSPAdes assemblies indicate that these assemblies contain a lower proportion of small contigs and therefore are less fragmented assemblies (1, 18, 21, 46).

Although the metaSPAdes assemblies were associated with 10 to 30% more CDSs than the MEGAHIT assemblies, the coding densities were similar across the assemblies of metaSPAdes and MEGAHIT and range between 0.77 and 0.80, indicating that none of the assembly strategies/assemblers were susceptible to disproportionately higher frameshift errors (Table S5). CDSs were blasted against the UniProtKB/TrEMBL nonredundant (nr) protein database to identify high-scoring segment pairs (HSPs), i.e., sequence pairs sharing high alignment scores, at an expected value (E value) cutoff of $1 \times 10^{-3}$. Across the assembly strategies, between 71 and 76% of the CDSs shared a high degree of similarity against the reference amino acid sequences in the UniProtKB/TrEMBL nr protein database and had average E values and bit scores of $3.09 \times 10^{-6} \pm 3.91 \times 10^{-5}$ and $387.00 \pm 305.65$, respectively (Table S5). Assembly fragmentation was assessed by analyzing the ratio between the lengths of CDSs (query length [qlen]) and their top hits in the UniProtKB/TrEMBL nr protein database (subject length [slen]), with higher qlen/slen ratios indicating less gene fragmentation and thus lower assembly fragmentation. Similar distributions in qlen/slen ratios were observed across the assemblies of metaSPAdes and MEGAHIT, with only between 30 and 36% of the CDSs having qlen/slen ratios ranging between 0.95 and 1 (Fig. S3A; Table S6.1). This suggests that the vast majority of CDSs across both assemblers and all assembly strategies were likely fragmented.

The CDSs of the metaSPAdes assemblies were less fragmented than those from the MEGAHIT assemblies and hence had higher mean qlen/slen ratios (Table S6.1; Fig. S3B).

*Post hoc* comparisons using the Tukey honestly significant difference (HSD) test indicated statistically significant differences between all metaSPAdes and MEGAHIT assemblies (Tukey HSD test, all $P < 0.05$) (Tables S6.2 and S6.3). metaSPAdes time-discrete assembly had the greatest mean qlen/slen ratio ($0.901 \pm 0.307$), followed by MASH distance-based assembly ($0.895 \pm 0.315$), individual assembly ($0.895 \pm 0.306$), and coassembly ($0.892 \pm 0.320$). Similar observations were made for the MEGAHIT assembly strategies (Table S6.1). Though significant differences were found, the effect sizes of these differences were small (effect size [$\eta^2$], 1.15E-04 and 2.39E-04 for metaSPAdes and MEGAHIT, respectively), suggesting that only 0.01 and 0.02% of the change in qlen/slen ratios can be accounted for by the assembly strategy for metaSPAdes and MEGAHIT, respectively. Statistically significant differences in the variation around the mean qlen/slen ratios were observed for the assemblies of metaSPAdes (coefficient of variation [CV], $0.35 \pm 0.01$) and MEGAHIT (C, $0.36 \pm 0.01$) (signed-likelihood ratio test [SLRT] and asymptomatic test, all $P < 0.05$) (Table S6.4). Associations between the mean qlen/slen ratios and CV estimates indicated that the metaSPAdes assembly strategies were associated with higher qlen/slen ratios and lower CV estimates than the assemblies of MEGAHIT (Fig. S3B). The ratio between the alignment lengths of CDSs (query alignment length [qalignlen]) and their top hits in the UniProtKB/TrEMBL nr protein database (subject alignment length [salignlen]) were used to evaluate potential misassembly due to the presence of insertion/deletion (indels) in genes (Table S7.1). Similar distributions in qalignlen/salignlen ratios were observed across the assemblies of metaSPAdes and MEGAHIT, with between 59 and 66% of the CDSs having qalignlen/salignlen ratios that ranged between 0.95 and 1 (Fig. S3C; Table S7.1). Though statistically significant differences in the mean qalignlen/salignlen ratios were observed for the assemblies of metaSPAdes and MEGAHIT (ANOVA, all $P < 0.05$) (Tables S7.2 and S7.3), the effect size of the differences was small ($\eta^2 = 6.22 \times 10^{-5}$ and $3.74 \times 10^{-5}$ for metaSPAdes and MEGAHIT, respectively). Similarly, small but statistically significant differences in the variation around the mean qalignlen/salignlen ratios of the metaSPAdes and MEGAHIT assembly strategies were observed (CV range, 0.02 to 0.03 for metaSPAdes and MEGAHIT assemblies) (SLRT and asymptomatic test, all $P < 0.05$) (Table S7.4). Overall, these results suggest that while metaSPAdes results in significantly less fragmented assemblies with lower rates of genes fragmentation, the effect size of this difference on CDS quality is small.

The proportion of sequencing information retained following assembly was determined by mapping the quality-trimmed paired-end reads of each sample to the nonredundant filtered metaSPAdes and MEGAHIT assemblies. Although a significant proportion of the assembled contigs ~98.39% were removed due to insufficient contig lengths, the majority of sequencing data were retained in contigs >1 kbp (Table S5). No statistically significant differences between the mean read mapping rates of corresponding metaSPAdes and MEGAHIT assembly strategies were observed (Tukey HSD, all $P < 0.05$) (Table S8); however, the metaSPAdes assemblies had mean mapping rates that were higher than the MEGAHIT assemblies (Fig. 1C; Table S5). Among the metaSPAdes assemblies, coassembly of all samples had the highest mapping rate ($90.61 \pm 3.28\%$), followed by time-discrete assembly ($88.45 \pm 3.64\%$), MASH distance-based assembly ($86.91 \pm 3.39\%$), and individual assembly ($85.47 \pm 4.45\%$). Similar observations were made for the MEGAHIT assemblies. Overall, the metaSPAdes assemblies had larger and more contiguous assemblies with read mapping rates of >85%. This confirms previous findings (1, 18, 21, 46).

**Evaluation of binning results for the combination of assembly strategies, assemblers, and binning approaches.** Unrefined bin sets were generated from the metaSPAdes and MEGAHIT assemblies of each assembly strategy using original binning algorithms that combine tetranucleotide frequencies and coverage information across samples (12, 13, 47), i.e., CONCOCT v1.1.1, MetaBAT v2.12.1, and MaxBin v2.2.4, as well as DAS Tool v1.1.0, which integrates results of bin predictions made by original binning algorithms to optimize the selection on nonredundant, high-quality bin sets (20). This resulted in 64 assembly/binning combinations ($n = 32$ assembly/binning combinations for bin sets that were constructed using larger than 1-kbp and 2.5-kbp

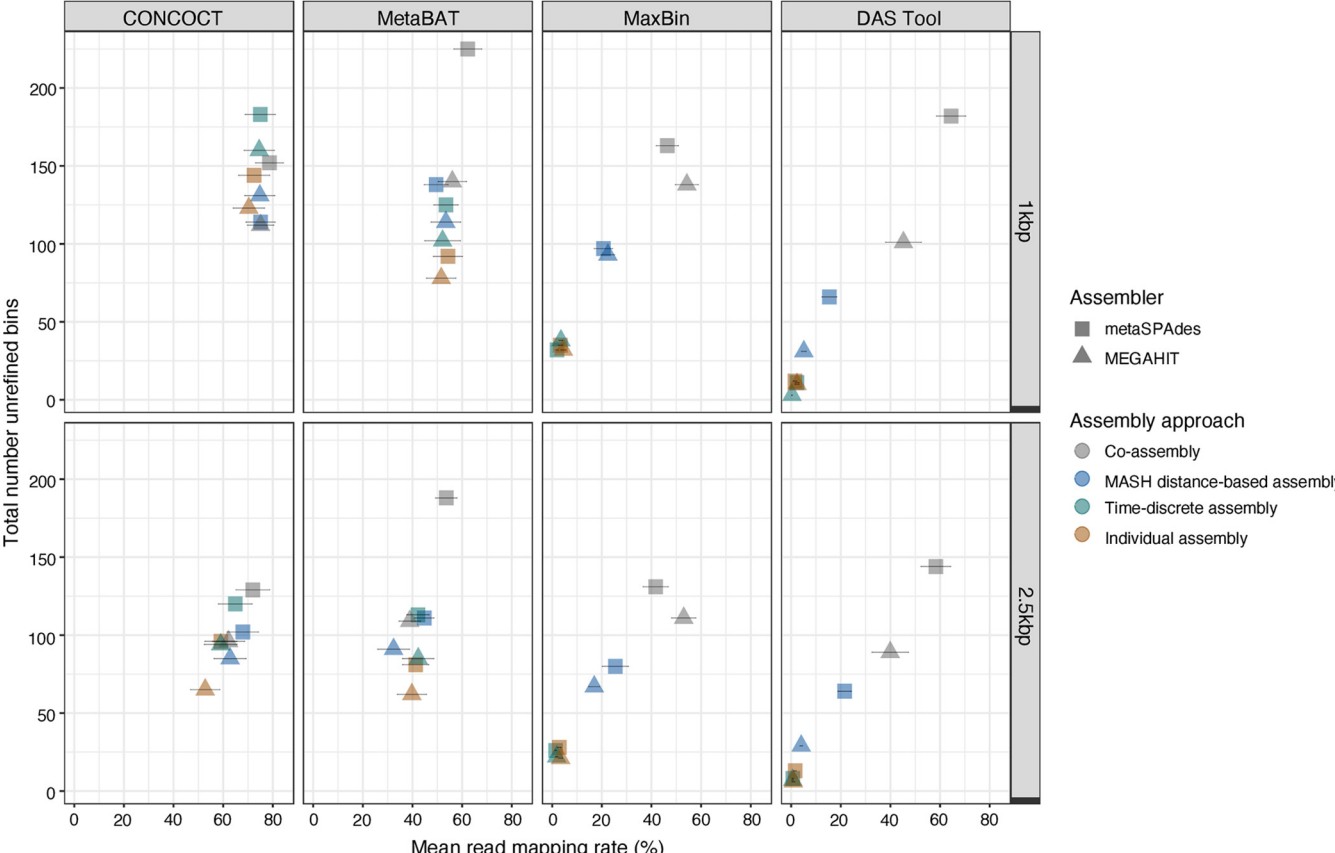

**FIG 2** Association between total number of bins and mean read mapping rates of sample reads mapped against the unrefined bins with completeness ≥50% that were assembled with different assembly approaches (coassembly [gray], MASH distance-based assembly [blue], time-discrete assembly [green], and individual assembly [orange]) using metaSPAdes (▲) and MEGAHIT (■) and binned with CONCOCT, MetaBAT2, MaxBin2, and DAS Tool. Error bars indicate standard errors of read mapping rates.

contigs, respectively) (Table S9). MASH distance-based nonmetric multidimensional scaling (NMDS) clustering of the unrefined bin sets indicated that the bin sets clustered based on assembly/binning approach rather than contig size cutoff used for binning (Fig. S4). The importance of assembly/binning approach compared to contig size cutoff was further confirmed by permutational analysis of variance (PERMANOVA) (Table S10). The minimum contig threshold (1 kbp or 2.5 kbp) that was selected for binning explained a smaller proportion of the variation, ~2% [PERMANOVA, $F(1) = 36.16$, $R^2 = 0.02$, $P < 0.05$] compared to 96% of the variation that was explained by the assembly/binning approach choice [PERMANOVA, $F(31) = 46.24$, $R^2 = 0.96$, $P < 0.05$]. However, the unrefined bin sets that were generated using contigs >1 kbp produced about 20% more unrefined bins with ≥ 50% completeness than the unrefined bin sets that were generated using contigs >2.5 kbp (Fig. 2; Table S9). These unrefined bins were, furthermore, associated with mapping rates that were between 5 and 20% higher than the unrefined bins that were generated using contigs >2.5 kbp (Fig. 2; Table S9). These findings suggest that binning with contigs >1 kbp allows for a more accurate representation of the microbial diversity.

**Marked improvements in bin qualities following reassembly and curation.** Across the 64 assembly/binning combinations greater than 1-kbp and 2.5-kbp contig size bin sets of 4 assembly/binning approaches, hence, 8 assembly/binning combinations in total, consistently produced the highest number bins (completeness ≥ 50%) and mapping rates greater than 50%. These assembly/binning approaches included the following coassembly strategies of metaSPAdes: (i) metaSPAdes coassembly and CONCOCT, (ii) metaSPAdes time-discrete assembly and CONCOCT, (iii) metaSPAdes

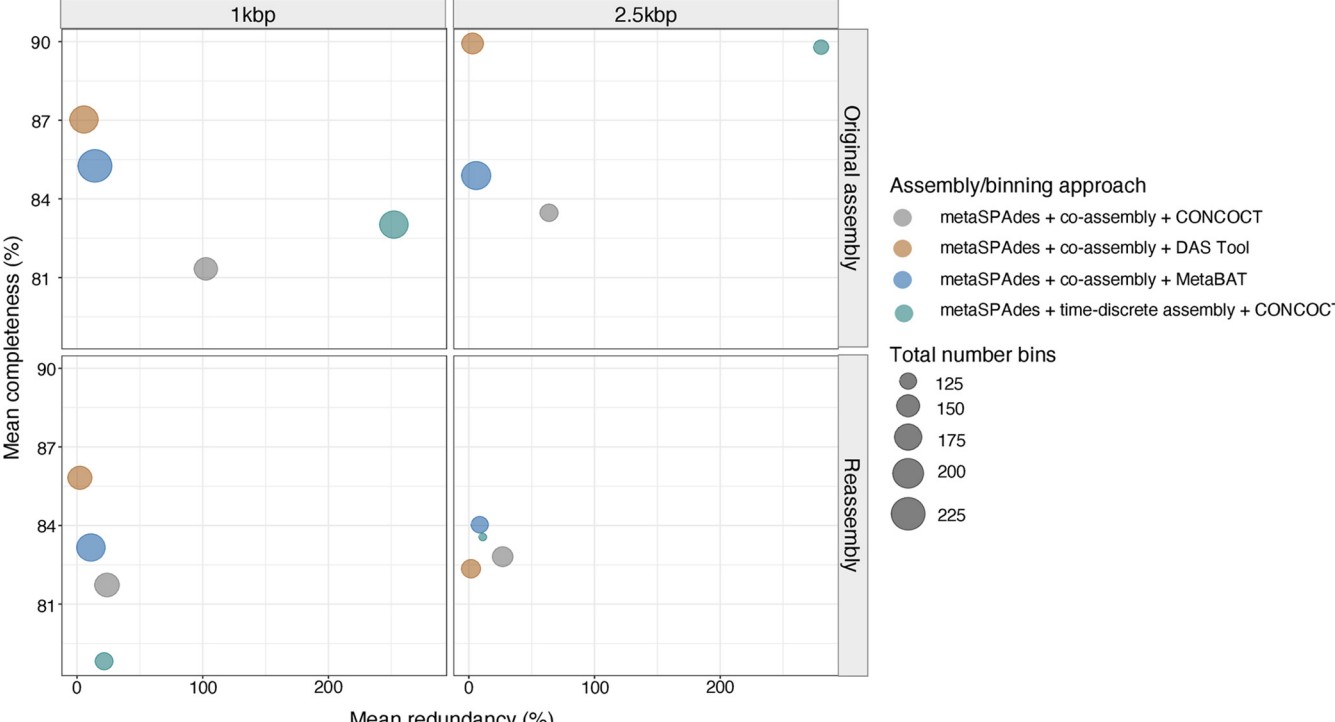

**FIG 3** Bubble plot showing the total number of bins (depicted by size) with mean completeness and redundancy estimates of unrefined bin sets (completeness estimates > 50%) that were generated before and after reassembly using 4 assembly/binning approaches (metaSPAdes coassembly and CONCOCT [orange], metaSPAdes coassembly and DAS Tool [gray], metaSPAdes coassembly and MetaBAT2 [blue], and metaSPAdes time-discrete assembly and CONCOCT [green]) and binning of contigs greater than 1 kbp and 2.5 kbp.

coassembly and MetaBAT2, and (iv) metaSPAdes coassembly and DAS Tool (Fig. 2). The unrefined CONCOCT bin sets of the coassembled metaSPAdes assemblies (i.e., metaSPAdes coassembly and CONCOCT and metaSPAdes time-discrete assembly and CONCOCT) consisted of fewer bins and bins that were significantly greater in size (Table S11). In particular, for metaSPAdes coassembly and CONCOCT, the average bin size was 8.15 ± 8.12 Mbp and 6.41 ± 5.26 Mbp for 1-kbp and 2.5-kbp constructed bins, while metaSPAdes time-discrete assembly and CONCOCT had average bin sizes of 14.05 ± 15.06 Mbp (contigs > 1 kbp) and 15.01 ± 12.59 Mbp (contigs > 2.5 kbp). These bins were also associated with large redundancy estimates that averaged above 60% and average strain heterogeneity estimates greater than 20% (Fig. 3). These findings suggest that the unrefined CONCOCT bin sets likely consist of multigenome or chimeric bins and highlighted the need for reassembly of individual bins and/or bin curation (19). The remaining coassembly strategies of metaSPAdes, i.e., metaSPAdes coassembly with MetaBAT2 and metaSPAdes coassembly with DAS Tool, generated more bins and bins with lower redundancy estimates that average below 15% (Fig. 3). To improve the quality of the unrefined bin sets, bins with greater than 50% completeness of the 8 assembly/binning combinations were independently subjected to reassembly. Proper paired quality-trimmed reads associated with the bins were extracted, converted to FASTQ format, and then reassembled using metaSPAdes and rebinned using the appropriate original binning approach. Following reassembly, the reassembled unrefined bin sets of metaSPAdes coassembly with CONCOCT and metaSPAdes time-discrete assembly with CONCOCT consisted of bins that were notably smaller in size (Table S11). Specifically, the reassembled metaSPAdes time-discrete assembly and CONCOCT had average unrefined bin sizes for the 1-kbp- and 2.5-kbp constructed bins (4.21 ± 2.75 Mbp and 3.97 ± 1.66 Mbp, respectively) that were at least 4-fold lower than the original unrefined bin sizes. Similar observations were made for reassembled metaSPAdes coassembly and CONCOCT that had an average 2-fold reduction in bin size (Table S11). Furthermore,

reductions in bin size across the coassembled CONCOCT bins that were reassembled was accompanied by improvements in bin quality (Fig. 3). These improvements were associated with reduced redundancy estimates across the 1-kbp and 2.5-kbp unrefined bin sets of reassembled metaSPAdes coassembly and CONCOCT (26.91 ± 47.05% and 23.86 ± 42.87%) and reassembled metaSPAdes time-discrete assembly and CONCOCT (21.52 ± 41.98% and 11.08 ± 27.92%). These findings suggest that the unrefined CONCOCT bins set consisted of chimeric bins that were resolved with reassembly. This improvement in bin quality after reassembly is consistent with previous findings (48). In contrast, no improvements in bin quality were observed in the reassembled bin sets of MetaBAT2 and DAS Tool (Fig. 3). Specifically, the reassembled unrefined 1-kbp and 2.5-kbp bin sets of metaSPAdes coassembly and MetaBAT2 maintained smaller bin sizes (4.16 ± 3.04% and 4.28 ± 2.66%) as well as average redundancy estimates below approximately 10% (11.05 ± 29.38% and 8.66 ± 20.47%). Similar observations were made for metaSPAdes coassembly and DAS Tool. This was expected, as higher-quality bins were associated with both MetaBAT and DAS Tool bin sets prior to reassembly.

**Metagenome-assembled genomes shared across assembly/binning approaches.** The reassembled CONCOCT bin sets (i.e., reassembled metaSPAdes coassembly with CONCOCT and reassembled metaSPAdes time-discrete assembly with CONCOCT) and original assembled MetaBAT and DAS Tool bin sets were manually curated using the interactive interface of anvi'o v6.1 (49) to obtain final MAGs (completeness ≥ 50% and redundancy < 10%) (Fig. S5). In total 1,279 MAGs were generated across the 4 assembly/binning approaches that were constructed using contigs >1 kbp ($n = 673$) and contigs >2.5 kbp ($n = 606$). Approximately 98% ($n = 1,259$) of the MAGs that were identified met the metagenome-assembled genome (MIMAG) standard (50) for medium-quality draft genomes, while only 20 MAGs were classified as high-quality draft genomes (Table S12.1). The limited number of high-quality MAGs was mainly due to the absence of a full complement of rRNA genes. Depending on the assembly/binning approach, between 75 and 83% of the MAGs lacked 16S rRNA genes, while between 10 and 16% of the MAGs consisted of fragmented 16S rRNA gene(s) (Table S12.1). MAGs often lack 16S rRNA genes due to their conserved and repetitive nature, which results in fragmented assemblies (24, 51, 52). Overall, none of the assembly/binning strategies produced sufficient high-quality MAGs as defined under MIMAG standards. Alternative sequencing technologies (e.g., long read) may be able to successfully reconstruct full complementary rRNA genes to increase the number of high-quality MAGs (5).

The MAGs across the 4 assembly/binning approaches shared similar characteristics in terms of contiguity (i.e., total length and $N_{50}$) and quality (i.e., completeness, redundancy, and strain heterogeneity) (Fig. 4A and Table S12.1). Overall, the curated MAG sets of the assembly/binning approaches retained more than 66% of the sequencing information (Table S12.1). The curated MAG sets of DAS Tool and MetaBAT had higher mapping rates, ~70%, than the curated reassembled MAG sets of CONCOCT. As shown in Fig. 4B, MAGs that were reconstructed using a minimum contig length of 1 kbp had slightly higher mapping rates than the mapping rates of the MAGs that were reconstructed using a minimum contig length of 2.5 kbp. These differences in read mapping rates were not statistically significantly different between corresponding assembly/binning strategies that used minimum contig threshold of 1 kbp and 2.5 kbp for MAG reconstruction, respectively (Tukey HSD test, all $P > 0.05$) (Table S12.2).

The curated MAG sets clustered by assembly/binning approach based on MASH distance estimates that explained approximately 91% of the variation in the nucleotide composition [PERMANOVA, $F(3) = 16.80$, $R^2 = 0.91$, $P < 0.05$] (Fig. 4C; Table S13). Though the minimum contig threshold (1 kbp or 2.5 kbp) that was selected for binning explained a smaller proportion of the variation of ~3%, this was not significant [PERMANOVA, $F(1) = 1.73$, $R^2 = 0.03$, $P > 0.05$] (Table S13). Based on MASH distance estimates, the differences in nucleotide composition of the curated MAGs between assembly/binning approaches were small, ranging between 0.005 and 0.08. Reassembled metaSPAdes

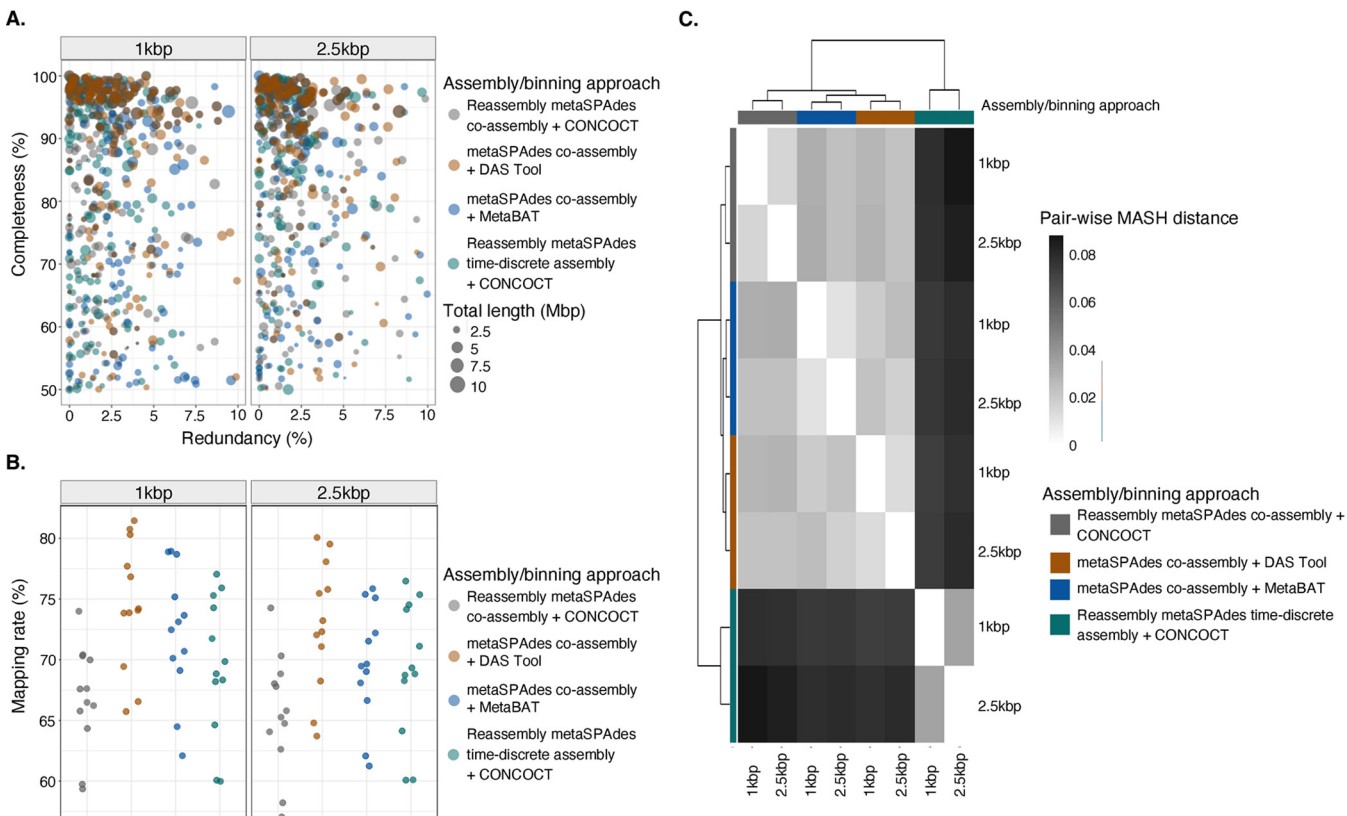

**FIG 4** Summary statistics and characteristics of 1,279 curated MAGs that were generated across the 4 assembly/binning approaches (reassembled metaSPAdes coassembly and CONCOCT [gray], metaSPAdes coassembly and DAS Tool [orange], metaSPAdes coassembly and MetaBAT2 [blue], and reassembled metaSPAdes time-discrete assembly and CONCOCT [green]) using contigs larger than 1 kbp (n = 673) and 2.5 kbp (n = 606), respectively. (A) Bubble plot showing total MAG size (depicted by size) and completeness (x axis) and redundancy (y axis) estimates of 1,279 curated MAGs that were generated for each of the 4 assembly/binning approaches. (B) Proportion of reads of 12 drinking water samples (●) that were mapped against the curated MAGs of each assembly/binning approach. (C) Comparison of the curated MAGs' nucleotide composition across the different assembly/binning approaches according to MASH distance. The heatmap are colored according to MASH distance; white denotes a distance of 0. Labels on the x and y axes are colored according to assembly/binning approach, and clustering was done using Euclidean distance. For a complete list of continuity and quality estimates, please refer to Table S12 in the supplemental material.

time-discrete assembly clustered separately from the other assembly/binning strategies, suggesting differences in the nucleotide composition of these curated MAGs.

Similarities in the nucleotide composition of the curated MAG sets and comparable MAG characteristics (i.e., continuity and quality) suggest the presence of overlapping MAGs across the assembly/binning approaches that likely represent the same species. The presence of overlapping MAGs across the assembly/binning approaches was investigated by aggregating all the MAGs and then clustering them using a 95% average nucleotide identity (ANI) threshold to identify species-level representative genomes (SRGs). Although the species concept for prokaryotes is controversial, this operational definition is commonly used and considered a golden standard (53). A total of 233 SRGs with average dRep quality scores {calculated as $(A \times$ completeness$) - (B \times$ contamination$) + C \times$ [contamination $\times$ (strain_heterogeneity/100)] $+ D \times \log(N_{50}) + E \times \log($size$) + F \times ($centrality $-$ S_ani)} (54) of 74.40 $\pm$ 21.80% were identified across the assembly/binning approaches (Table S14). These SRGs had average sizes of 3.46 $\pm$ 1.72% and were nearly complete (81.98 $\pm$ 16.39%), with redundancy estimates less than 10%. Taxonomic classification of the SRGs using GTDB-Tk classified 33 SRGs to species level, 178 to genus level, 217 to family level, and 233 to order, class, and phylum levels (Fig. 5A and Table S14).

Approximately 34% (n = 79) of the SRGs were shared across the assembly/binning approaches, where they accounted for between 39 and 48% of the sequencing data (Fig. 5B and Table S15). These SRGs had better quality with average completeness and redundancy estimates of 94.03 $\pm$ 8.3% and 1.3 $\pm$ 1.17%, respectively. Unique SRGs

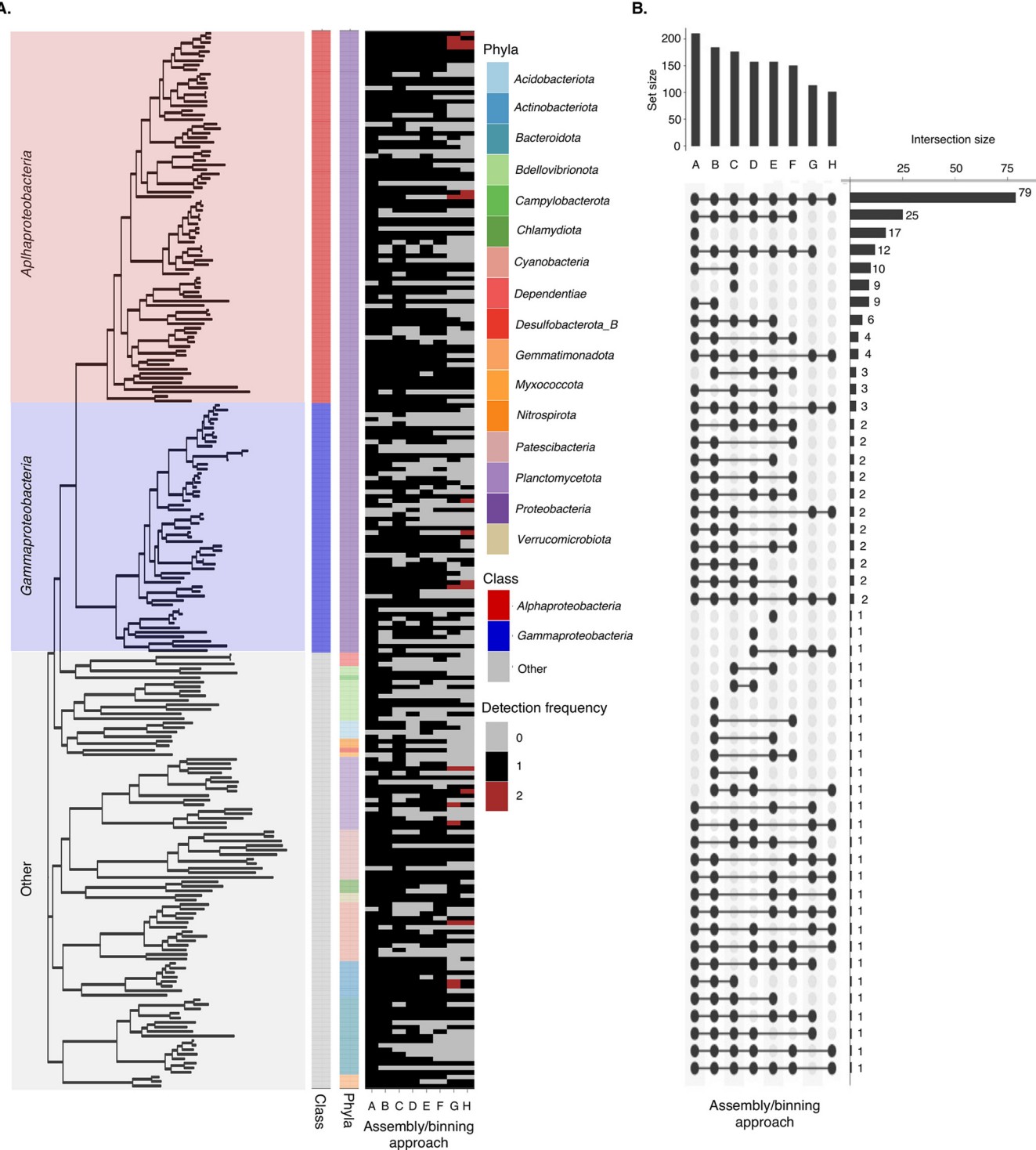

**FIG 5** (A) Phylogenomic analysis of 233 nearly complete SRGs inferred from 37 single-copy ribosomal bacterial core genes. The two inner panels represent taxonomic classification of 16 bacterial phyla and class-level classification of dominating bacterial phyla, *Proteobacteria* representing *Alphaproteobacteria* (red) and *Gammaproteobacteria* (blue). Outer panel represents a presence/absence summary plot showing the frequency distribution of MAGs that were constructed using contigs greater than 1 kbp and 2.5 kbp across 4 assembly/binning approaches. Gray denotes absence, while black and red denotes the presence of a singular MAG or duplicate MAGs that demonstrated ≥95% ANI, respectively. (B) UpSetR plot showing the distribution of species-level representative genomes (SRGs) that demonstrated ≥95% ANI between the 4 assembly/binning approaches in which MAGs were constructed using contigs greater than 1 kbp and 2.5 kbp. For the assembly/binning approaches, A represents metaSPAdes coassembly and MetaBAT2 (1 kbp), B indicates metaSPAdes coassembly and MetaBAT2 (2.5 kbp), C represents metaSPAdes coassembly and DAS Tool (1 kbp), D represents metaSPAdes coassembly and DAS Tool (2.5 kbp), E represents reassembled metaSPAdes coassembly and CONCOCT (1 kbp), F represents reassembled metaSPAdes coassembly and CONCOCT (2.5 kbp), G represents reassembled metaSPAdes time-discrete assembly and CONCOCT (1 kbp), and H represents reassembled metaSPAdes time-discrete assembly and CONCOCT (2.5 kbp).

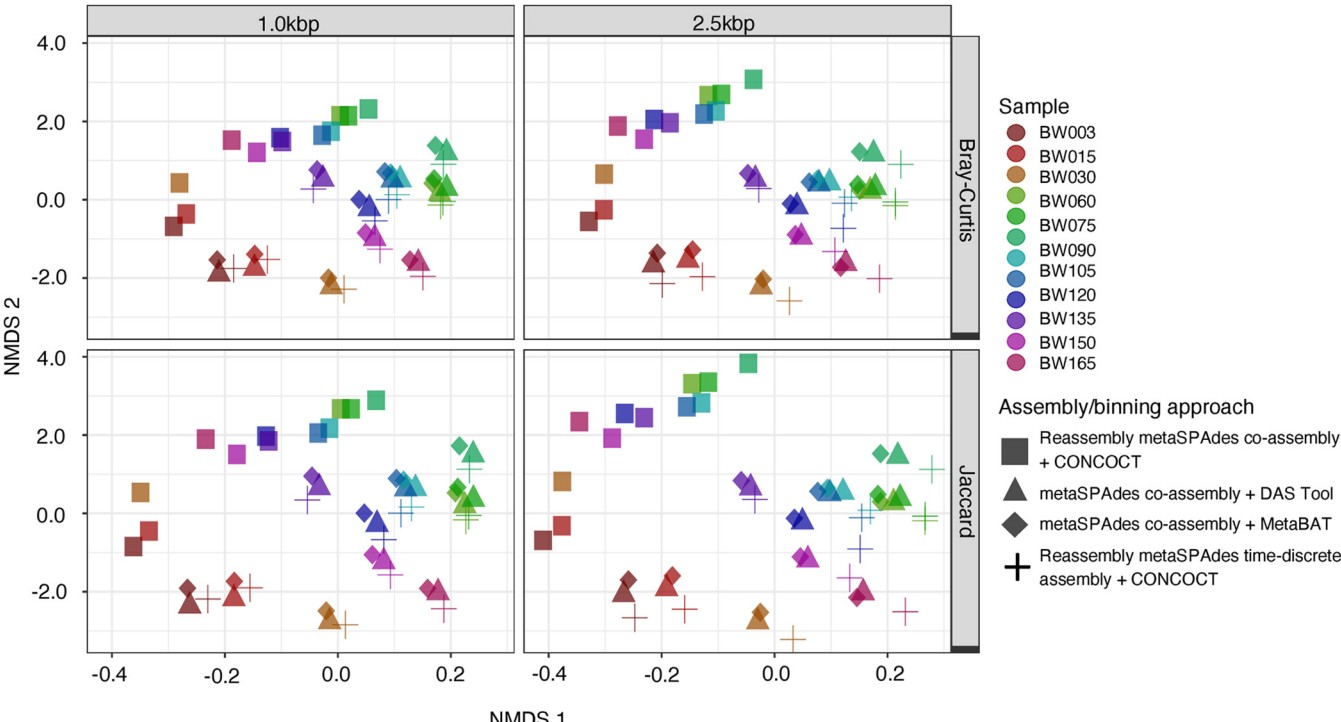

**FIG 6** Structure-based nonmetric multidimensional scaling (NMDS) of Bray-Curtis dissimilarity matrices and membership-based Jaccard distances as inferred using the abundance information (in reads per kilobase per million [RPKM]) of all MAGs identified across the assembly/binning approaches. Points represent samples (or time points) (see Table S1 in the supplemental material), and shapes represent the assembly/binning approaches, i.e., ■, reassembled metaSPAdes coassembly and CONCOCT; ▲, metaSPAdes coassembly and DAS Tool; ♦, metaSPAdes coassembly and MetaBAT2; and +, reassembled metaSPAdes time-discrete assembly and CONCOCT.

represented 10% ($n = 29$) of the total SRGs, while the largest proportion of the SRGs, ~43% ($n = 125$), were shared between two or more assembly/binning approaches (but not all) (Fig. 5B). The latter shared similar quality characteristics to the SRGs that were shared across all the assembly/binning approaches and accounted for between 18 and 32% of the sequencing data. Overall, metaSPAdes coassembly and MetaBAT2 (with contigs > 1 kbp) retained more SRGs ($n > 200$) and were able to reconstruct MAGs that were not detected in the other assembly/binning approaches (Fig. 5A). Though reassembled metaSPAdes time-discrete assembly and CONCOCT were associated with a low number of SRGs ($n < 120$), 12 duplicate SRGs sharing >95% ANI were identified within the 1-kbp and 2.5-kbp approaches, respectively. These SRGs were likely subspecies, suggesting that the reassembled metaSPAdes time-discrete assembly and CONCOCT assembly/binning approach can differentiate between closely related species that were otherwise collapsed or considered a singular strain using the other assembly/binning approaches. This highlights the potential for utilizing multiple approaches for not just binning but also assembly strategies, as this can assist in the recovery of a greater proportion of populations in metagenomes.

Reads from all samples were mapped to the 233 SRGs and their relative abundance in each sample for all assembly/binning, and contig size strategy was estimated based on the presence/absence of the bin in the respective strategy. Variability in the microbial community structure and membership between the assembly/binning approaches were visualized by ordinating the samples in multidimensional space. As shown in Fig. 6, the samples cluster by time point which explained approximately 64% of the variation in the community membership [PERMANOVA, $F(11) = 108.88$, $R^2 = 0.637$, $P < 0.05$] and 70% of the variation in community structure [PERMANOVA, $F(11) = 50.92$, $R^2 = 0.703$, $P < 0.05$] (Tables S16.1 and S16.2). The remaining variables, i.e., assembly/binning approach and contig size, accounted for a smaller but significant proportion of the variation. In particular, the assembly/binning approach explained about 29% of the variation in community

Microbiology
Spectrum

membership [PERMANOVA, $F(3) = 182.78$, $R^2 = 0.291$, $P < 0.05$], while contig size explained 3% [PERMANOVA, $F(3) = 55.50$, $R^2 = 0.03$, $P < 0.05$]. Similar observations were made for structure-based analysis (Table S16.1). Although no clustering by assembly/binning approach and contig size was observed, the average dissimilarity in community structure and membership between time points were about 37 and 53%, respectively ($d_{BC}$ and $d_J = 0.369 \pm 0.11$ and $0.528 \pm 0.13$). These findings suggest that while temporal dynamics of the drinking water microbiome are largely retained despite variation in genome-centric metagenomic workflows, the choice of assembly/binning strategy, in particular, can have a significant impact on the structure and membership of the drinking water microbiome and should not be overlooked.

**Conclusion.** This study evaluated the performance of a combination of *de novo* assembly strategies and binning algorithms for time series metagenomic data for drinking water microbial communities to identify an ideal combination of assembly and binning approaches that allow for the generation of high-quality metagenomic assemblies and MAGs. Overall, metaSPAdes coassembly strategies, i.e., coassembly of all samples and time discrete assembly, produced less fragmented and larger assemblies that retained the maximum amount of metagenomic information. Reassembly and binning followed by manual curation significantly improved MAG qualities in situations with unresolved multigenome or chimeric bins. Though none of the assembly/binning strategies were able to reconstruct high-quality MAGs due to the absence of a full complement of rRNA genes, metaSPAdes coassembly and MetaBAT2 retained the highest number of medium-quality MAGs and were able to reconstruct MAGs that were not detected with the other assembly/binning approaches. Moreover, reassembled metaSPAdes time-discrete assembly and CONCOCT were able to differentiate between closely related species that were otherwise collapsed or considered a singular strain using the other two assembly/binning approaches. Our study also finds that the choice of assembly/binning strategy can have a significant impact on the membership and structure of the microbial community as inferred from presence/absence and relative abundance of MAGs. This, combined with the fact that a significant proportion of SRGs were not reconstructed using any single approach, highlights the need to utilize multiple assembly/binning approaches for MAG recovery. We therefore recommend utilizing multiple assembly, binning, and binning-aggregating strategies followed by dereplication to maximize the recovery of nonredundant MAGs that may more fully represent the microbial populations in drinking water samples.

## MATERIALS AND METHODS

**Sample collection.** A drinking water sample was collected every second week over a period of 6 months from a tap in a commercial building located in Boston, Massachusetts (United States). This resulted in 12 drinking water samples that represented successive sampling points (Table S1 in the supplemental material). Prior to sample collection, the system was flushed for at least 30 min at a flow rate ranging between 3.0 and 3.3 l.min$^{-1}$, and then approximately 1,500 ml of tap water was collected for microbial community analysis in a sterile (by autoclaving) a 2-liter Duran GLS 80 wide-mouth borosilicated glass bottle (Duran; catalog no. 1112715). An additional 500-ml sample was collected in parallel in sterile 2 × 250 ml Duran GLS 80 wide-mouth borosilicated glass bottles (Duran; catalog no. 218603656) for chemical analysis. Samples for microbial community analysis were filtered immediately through Sterivex-GP pressure filter units (EMD Millipore; catalog no. SVGP01050) containing a 0.22-$\mu$m polyethersulfone (PES) filter membrane, using the Geotech Geopump series II peristaltic pump (Geotech Environmental Equipment, Inc.; catalog no. 91350113) and sterile size 15 Geotech silicone tubing (Geotech Environmental Equipment, Inc.; catalog no. 77050000). Following filtration, the exterior of the filter unit was cleaned with a 70% ethanol (Fisher Scientific; catalog no. A962F)-soaked Kimwipe (Kimberly Clark Professional; catalog no. 34120) and then transferred to a 50-ml Falcon tube (Corning; catalog no. 362070) and stored at $-80°C$ until further analysis.

**Water chemistry characterization.** Water quality parameters (i.e., temperature, pH, conductivity, and dissolved oxygen) were measured using the Orion Star A325 pH/conductivity portable multiparameter meter (Thermo Scientific; catalog no. STARA3250). Total chlorine was measured using U.S. Environmental Protection Agency (U.S. EPA)-approved Hach method 8167 with DPD total chlorine reagent powder pillows (Hach; catalog no. 2105669). Reactive orthophosphate was measured using U.S. EPA-approved Hach method 8048 with PhosVer 3 phosphate reagent powder pillows (Hach; catalog no. 2106028). Nitrogen species, including ammonium, nitrate, and nitrite were measured using the nitrogen-ammonia reagent set (method 10023; Hach; catalog no. 2604545), NitraVer X nitrogen-nitrate reagent set (method 10020; Hach; catalog no. 2605345), and NitriVer 3 TNT reagent set (method 10019;

catalog no. 2608345), respectively. All Hach measurements were performed in triplicate on the DR1900 portable spectrophotometer (Hach; catalog no. DR190001H) (Table S1).

**Flow cytometric analysis.** Standard flow cytometric measurements (FCM) were performed as described previously (30, 55). Briefly, samples were quenched with 10 mM sodium thiosulfate (1% [vol/vol]) (Alfa Aesar; catalog no. AA35645K2) and then preheated at 37°C for 3 min, stained with SYBR green I (SG) (Invitrogen; catalog no. S7585) (1:100 diluted in 10 mM Tris-HCl, pH 8.5; Bioworld; catalog no. NC1213695) at 10 $\mu$l·ml$^{-1}$ or SG combined with propidium iodide (PI) (Molecular Probes; catalog no. P3566) (3 $\mu$M final concentration) at 12 $\mu$l·ml$^{-1}$ and incubated in the dark at 37°C for 10 min. Five negative controls consisting of (i) unstained UltraPure DNase/RNase-free distilled water (Thermo Fisher Scientific; catalog no. 10977015), (ii) SG-stained UltraPure DNase/RNase-free distilled water, (iii) SGPI-stained UltraPure DNase/RNase-free distilled water, (iv) SG-stained 0.22-$\mu$m filtered tap water sample, and (v) SGPI-stained 0.22-$\mu$m filtered tap water sample, respectively, were processed identically and in parallel with the samples. FCM was performed on a 50-$\mu$l sample in triplicate at a preset flow rate of 66 $\mu$l·min$^{-1}$ using a BD Accuri C6 flow cytometer (BD Accuri cytometers, Belgium), which is equipped with a 50-mW solid-state laser emitting light at a fixed wavelength of 488 nm. Green and red fluorescent intensities were collected at FL1 of 533 $\pm$ 30 nm and FL3 of >670 nm, respectively, along with sideward- and forward-scatter light intensities. Data were processed with the BD Accuri CFlow software that permits electronic gating to separate the positive signals from instrumental and sample background noise on a two-parameter density plot (30). A trigger/threshold of 1,000 was applied on the green fluorescence channel (FL1). No compensation was used.

**Sample processing and DNA extraction.** Prior to extraction, the bead constituents (i.e., ceramic and silica spheres and glass bed) contained within the 2-ml Lysing Matrix E tubes (MP Biomedicals; catalog no. 116914100) were aseptically transferred into sterile 1.5-ml microcentrifuge tubes (Eppendorf; catalog no. 022431021) (56). Removal of these components was necessary to ensure that the processed PES filter membranes from the Sterivex-GP pressure filter units are fully immersed in solution during the enzymatic and chemical treatment steps of the DNA extraction protocol. The PES filter membrane with harvested microbial biomass was aseptically removed from the Sterivex-GP pressure filter unit and cut into smaller pieces on the surface of a petri dish (Fisher Scientific; catalog no. FB0875712) using a sterile scalpel (Fisher Scientific; catalog no. 08-920B) and then transferred into the emptied 2-ml Lysing Matrix E tubes using a sterile tweezer (Fisher Scientific; catalog no. 22-327379). DNA extractions were performed using a modified version of the DNeasy PowerWater kit (Qiagen; catalog no. 14900-50-NF or 14900-100-NF) protocol that utilizes enzymatic, chemical, and mechanical lysis strategies to enhance recovery of DNA from drinking water samples (56). Briefly, filter cuttings contained in the 2 ml Lysing Matrix E tubes were submerged in 294 $\mu$l 10$\times$ Tris-EDTA (100 mM Tris, 10 mM EDTA, pH 8.0; G-Biosciences; catalog no. 501035446) and 6 $\mu$l lysozyme solution (50 mg·ml$^{-1}$; Thermo Fisher Scientific; catalog no. 90082) and incubated for 60 min at 37°C with light mixing at 300 rpm using the Eppendorf ThermoMixer C (Eppendorf; catalog no. 2231000680). Subsequently, the tubes were supplemented with 300 $\mu$l prewarmed (55°C) PW1 solution, provided with the DNeasy PowerWater kit, and 30 $\mu$l proteinase K (20 mg·ml$^{-1}$; Thermo Fisher Scientific; catalog no. AM2546), vortexed, and incubated for 30 min at 56°C with light mixing at 300 rpm using the Eppendorf ThermoMixer C. After incubation, the bead constituents initially transferred to the sterile 1.5-ml microcentrifuge tubes were aseptically transferred back to the Lysing Matrix E tubes. The tubes were then supplemented with 630 $\mu$l chloroform/isoamyl alcohol (24:1, pH 8; Acros Organics; catalog no. 327155000) and beads beaten at setting 6 for 40 s using the FastPrep-24 classic instrument (MP Biomedicals; catalog no. 116004500). The resulting homogenized mixture was then subjected to centrifugation at 14,000 $\times$ $g$ for 10 min at 4°C using the Eppendorf centrifuge 5424R (catalog no. 5404000332). After centrifugation, the aqueous phase (600 to 650 $\mu$l) was transferred to a sterile 1.5-ml microcentrifuge tube. Exactly 600 $\mu$l of the aqueous phase was used as starting material on the QIACube system (Qiagen; catalog no. 9001882) to purify DNA according to the manufacturer's instructions using the DNeasy PowerWater kit protocol. Three negative controls consisting of a reagent blank (C01) and two filter blanks (i.e., unused PES membrane filters [C02] and PES membrane filters treated with autoclave deionized water [C03]) were processed identically and in parallel with the samples. The extracted DNA was quantified in duplicate using the Qubit dsDNA high-sensitivity (HS) assay kit (Thermo Fisher Scientific; catalog no. Q32851) with the Qubit 4 fluorometer (Thermo Fisher Scientific; catalog no. Q33238) (Table S2). All DNA extracts (50 $\mu$l) were stored at $-80$°C until further analysis.

**Quantitative PCR.** The quantitative PCR (qPCR) assay was performed on a QuantStudio 3 real-time PCR system (Thermo Fisher Scientific; catalog no. A28567) in a 20-$\mu$l reaction mixture consisting of Luna Universal qPCR master mix (New England Biolabs, Inc.; catalog no. NC1276266), forward and reverse primer pairs (F515-GTGCCAGCMGCCGCGGTAA and R806-GGACTACHVGGGTWTCTAAT, respectively) (57), UltraPure DNase/RNase-free distilled water (Thermo Fisher Scientific), and 1:10 diluted DNA template. Reactions were prepared in triplicate in a 96-well optical plate using the epMotion M5073 automated liquid handling system (Eppendorf; catalog no. 5073000205D). qPCR conditions were as follows: 1 min at 95°C and then 40 cycles consisting of 15 s at 95°C, 15 s at 50°C, and 1 min at 72°C. A calibration curve with standards ranging from 102 to 108 copies of the 16S rRNA gene of *Nitrosomonas europaea* for total bacteria assay was generated. The calibration curve for 16S rRNA copies was linear ($R^2 = 0.997$) over 7 orders of magnitude with a high PCR efficiency (100%).

**Metagenomic sequencing.** Sequencing libraries were prepared using the Ovation Ultralow DNA-seq library preparation kit (NuGen; catalog no. 0344NB). Metagenomic sequencing was performed on one S prime (SP) lane of the NovaSeq 6000 sequencing system (Illumina) at the Roy J. Carver Biotechnology Centre at the University of Illinois Urbana-Champaign (UIUC) Sequencing Core (Champaign, IL, USA).

**Sequence processing.   (i) Preprocessing.** Processing of sequencing data was done using the workflow outlined in Fig. S1. Initial quality control of FASTQ files was performed using fastp v0.20.0 (58) with parameters –trim_poly_x, –qualified_quality_phred 20, –length_required 20. The UniVec_Core database from NCBI (ftp://ftp.ncbi.nih.gov/pub/UniVec/) was subsequently used to screen for contaminant sequences (e.g., phix sequencing control used as sequencing control and sequencing adapters) by mapping the reads from each sample against the UniVec_Core database using BWA-MEM v0.7.17 (59) and then filtering reads in proper pair and supplementary alignments using SAMtools v1.9 (60) with parameters -hbS -F2 -F2048. BAM files were subsequently sorted using the sort function of SAMtools v1.9, and then quality-filtered forward and reverse FASTQ files were extracted from sequence alignments in sorted BAM format using the bamtofastq function of bedtools v2.29.2 (https://bedtools.readthedocs.io/en/latest/). The quality-filtered FASTQ files were analyzed using Nonpareil v3.303 (42) in k-mer mode to estimate the coverage and to predict the number of sequences required to achieve "near-complete" coverage. Nonpareil curves were generated in R (61) using the function Nonpareil.set of Nonpareil v3.3.4.

**(ii) MASH distance and k-means clustering.** MASH v2.2.2 (62) was used to estimate read-based dissimilarity between samples using the quality-filtered FASTQ files. For this, forward and reverse quality-filtered FASTQ files of each sample were interleaved using interleafq v1.0 (https://github.com/quadram-institute-bioscience/interleafq), and then the sketch function was used to convert the interleaved quality filtered FASTQ files of each sample into a MinHash sketch with parameters s = 100,000 and k = 21. The dist function was subsequently used for pairwise comparisons between samples based on Jaccard indices, thereby comparing the fraction shared k-mer between samples. K-means clustering on MASH distances was performed to partition samples into clusters with the nearest mean (Fig. S2). For this, the MASH distance matrix was imported into R, and the function fviz_nbclust of factoextra v1.0.7 (https://www.rdocumentation.org/packages/factoextra) was used to determine and visualize the optimal number of clusters (or k groups) using the average silhouette method with 999 Monte Carlo iterations. The MASH distance matrix was subsequently clustered by the k-means method using k-means of the stats package v3.6.2 (https://www.rdocumentation.org/packages/stats/versions/3.6.2/topics/kmeans) and then visualized using fviz_cluster of factoextra v1.0.7.

**(iii) Metagenomic assembly and binning.** The performance of a combination of assembly (metaSPAdes v3.13.1 [43] and MEGAHIT v1.2.9 [44]), binning (CONCOCT v1.1.1 [13], MetaBAT v2.12.1 [12], and MaxBin v2.2.4 [47]) and bin-aggregating software (DAS Tool v1.1.0 [20]) was evaluated using four assembly strategies, including individual assembly and three coassembly approaches, i.e., coassembly with all samples, MASH distance-based assembly, and time-discrete assembly. This resulted in 32 combinations of assembler, assembly strategy, and binning approaches (Table S2). For MASH distance-based assemblies, the following three coassemblies consisting of pooled samples that were identified using pairwise MASH dissimilarity indices and k-means clustering were identified: (i) BW003, BW015, BW030, and BW060; (ii) BW075, BW090, and BW105; and (iii) BW120, BW135, BW150, and BW165 (Table S2). Samples pooled and coassembled for time-discrete assembly consisted of the following 11 combinations representing paired samples from successive sampling points: (i) BW003 and BW015, (ii) BW015 and BW030, (iii) BW030 and BW045, (iv) BW045 and BW060, (v) BW060 and BW075, (vi) BW075 and BW090, (vii) BW090 and BW105, (viii) BW105 and BW120, (ix) BW120 and BW135, (x) BW135 and BW150, and (xi) BW150 and BW165. Control samples (i.e., C01, C02, and C03) were pooled and assembled independently in both MASH distance-based and time-discrete assembly strategies.

Quality-filtered forward and reverse FASTQ files of samples for the individual and three coassembly strategies were assembled using metaSPAdes v3.13.1 and MEGAHIT v1.2.9 with k-mer sizes of 21, 33, 55, and 77. Following assembly and prior to binning, contigs of the MASH distance-based coassemblies ($n = 4$), time-discrete coassemblies ($n = 12$), and individual assemblies ($n = 15$) were pooled within each strategy, resulting in 6 pooled assemblies and 8 assemblies in total (6 pooled assemblies and 2 coassemblies) that were used in downstream processing (Table S2). Contigs <1 kbp were filtered from all assemblies using seqtk (https://github.com/lh3/seqtk). This was followed by the removal of redundant contigs, i.e., duplicate and contained contigs, using the dedupe function of BBTools v38.76 (https://github.com/BioInfoTools/BBMap/blob/master/sh/dedupe.sh) for the pooled assemblies. QUAST v5.0.2 (63) was used to assess the quality of the processed assemblies with default parameters. Mapping rates were determined by mapping the quality-trimmed paired-end reads to each assembly using BWA-MEM v0.7.17 (59) and then filtering unmapped reads using the view function of SAMtools v1.9 (60) with parameters -hbS -F4. BAM files were subsequently sorted using the sort function of SAMtools v1.9, and then coverM v0.4.0 (https://github.com/wwood/CoverM) was used to calculate contig-wise coverage with the method flag set to count. Prokka v1.14.6 (64) was used to identify coding DNA sequences (CDSs) in the contigs and to translate these CDSs to protein-coding amino acid sequences. Coding density was calculated by dividing the total CDS length (in Mbp) by the total assembly length (in Mbp). The blastp workflow of DIAMOND v0.9.36 (65) was used to align the protein-coding amino acid sequences against the UniProt Knowledgebase (UniProtKB)/TrEMBL nonredundant (nr) protein database (https://www.uniprot.org/downloads) at an expected value (E value) cutoff of $1 \times 10^{-3}$ to identify high-scoring segment pairs (HSPs). Predicted protein-coding amino acid sequences that aligned with reference protein-coding amino acid sequences in the UniProtKB/TrEMBL nr protein database were used to compute query/subject length ratios and query/subject length alignment ratios. These query/subject length and query/subject length alignment ratios were used as a measure to assess the extent of assembly fragmentation and misassembly.

Binning of contigs greater than 1.0kbp and 2.5 kbp was performed using the analysis and visualization platform for omics data (anvi'o) v6.1 (49). In this workflow, bins were generated using binning algorithms that combine tetranucleotide frequencies and coverage information across samples, including CONCOCT v1.1.1 (13), MetaBAT v2.12.1 (12), and MaxBin v2.2.4 (47). Since different binning tools

reconstruct genomes at various levels of completeness, a bin aggregation software, i.e., DAS Tool v1.1.0 (20), was used to integrate the results of bin predictions made by CONCOCT, MetaBAT2, and MaxBin2 to optimize the selection on nonredundant, high-quality bin sets using default parameters. Recently published binning algorithms, including variational autoencoders for metagenomic binning (VAMB) (66) and SemiBin (67), were not considered in this study but are important alternatives to consider in the future to facilitate the recovery of high-quality MAGs. Bin statistics, including total size, number of contigs, $N_{50}$, GC content, etc., were obtained using the anvi-summarize function of anvi'o, while estimates of quality (completeness, redundancy, strain heterogeneity, etc.) were retrieved using the lineage-specific workflow of CheckM v1.0.18 (68). Mapping rates were determined by mapping the quality-trimmed paired-end reads to each bin using BWA-MEM v0.7.17, and then filtering unmapped reads using the view function of SAMtools v1.9 with parameters: -hbS -F4. BAM files were subsequently sorted using the sort function of SAMtools v1.9, and then coverM v0.4.0 was used to calculate contig-wise coverage with the method flag set to count.

To further improve bin quality, individual bins with ≥50% completeness of a selected combination of assembler, assembly strategy, and binning approach were identified for reassembly (see Results and Discussion). For this, properly paired quality-trimmed reads associated with individual bins of the selected assembly/binning approaches were extracted and stored into their FASTQ files using SAMtools v1.9 functions view and FASTQ, followed by assembly using metaSPAdes v3.13.1 with k-mer sizes 21, 33, 55, and 77. The reassembled contigs were rebinned a second time using the appropriate original binning approach, and bin statistics and mapping rates were determined as described above.

To obtain MAGs, bins were manually curated using the interactive interface of anvi'o v6.1. MAG characteristics and mapping rates were determined as described above. To assist in the identification of high- and medium-quality draft MAGs as defined under the minimum information about a metagenome-assembled genome (MIMAG) standards (50), ribosomal RNAs (rRNAs) and transfer RNAs (tRNAs) were detected with Prokka v1.14.6 (65). Pooled MAGs were dereplicated with dRep v2.6.2 (54) and clustered into species-level representative genomes (SRGs) at 95% average nucleotide identity (ANI). SRGs were classified using the classify workflow of the Genome Taxonomy Data set Toolkit (GTDB-Tk) v0.3.2 (69), which provides automated classification of bacterial genomes by placing them into domain-specific, concatenated protein reference trees. The phylogenomic workflow of anvi'o v6.1 was reproduced to construct a phylogenomic tree using a concatenated alignment of 37 single-copy ribosomal bacterial core genes.

**Statistical analysis.** Statistical analysis was performed in R (61). Descriptive statistics and statistics on central tendency were performed using one-way analysis of variance (ANOVA) provided in the stats package. Significant ANOVA findings were further investigated by performing a *post hoc* Tukey-Kramer test using the function Tukey.HSD with Bonferroni correction. Nonmultidimensional scaling (NMDS) using Bray-Curtis and Jaccard dissimilarity indices was performed using metaMDS provided in the vegan package, and permutational analysis of variance (PERMANOVA) was conducted using the function adonis of the vegan package. All plots were generated in R using ggplot2 (70).

**Data availability.** Raw sequence reads and 233 SRGs are available on NCBI at BioProject accession numbers PRJNA745168 and PRJNA745370, respectively. The 8 assemblies and 1,279 curated MAGs are available on datadryad at https://doi.org/10.5061/dryad.qnk98sfhr.

## SUPPLEMENTAL MATERIAL

Supplemental material is available online only.
**SUPPLEMENTAL FILE 1**, PDF file, 0.4 MB.
**SUPPLEMENTAL FILE 2**, XLSX file, 0.1 MB.

## ACKNOWLEDGMENT

This research was supported by NSF award number 1749530.

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
