## [Reviewer comments · Microbiology Spectrum]

Microbiology Spectrum

Evaluating de novo assembly and binning strategies for time-series drinking water metagenomes.

Solize Vosloo, Linxuan Huo, Christopher Anderson, Zihan Dai, Maria Sevillano, and Ameet Pinto

Corresponding Author(s): Ameet Pinto, Georgia Institute of Technology

Review Timeline:

Submission Date:

September 13, 2021

Accepted:

September 17, 2021

Editor: Jeffrey Gralnick

Reviewer(s): The reviewers have opted to remain anonymous.

Transaction Report:

DOI: <https://doi.org/10.1128/Spectrum.01434-21>

September 17, 2021

Dr. Ameet J Pinto
Georgia Institute of Technology
Civil and Environmental Engineering
Atlanta, GA 30318

Re: Spectrum01434-21 (Evaluating de novo assembly and binning strategies for time-series drinking water metagenomes.)

Dear Prof. Pinto:

Based on your thoughtful responses to the prior round of review and corresponding revisions, your manuscript has been accepted, and I am forwarding it to the ASM Journals Department for publication. You will be notified when your proofs are ready to be viewed.

Sincerely,

Jeffrey Gralnick
Editor, Microbiology Spectrum

Journals Department
Supplemental Tables: Accept
Supplemental Figures: Accept